# Air-Assisted Electrospinning of Dihydromyricetin-Loaded Dextran/Zein/Xylose Nanofibers and Effects of the Maillard Reaction on Fiber Properties

**DOI:** 10.3390/molecules29133136

**Published:** 2024-07-01

**Authors:** Yupeng Ren, Jianhui An, Cheng Tian, Longchen Shang, Yexing Tao, Lingli Deng

**Affiliations:** 1College of Biological and Food Engineering, Hubei Minzu University, Enshi 445000, China; 202230344@hbmzu.edu.cn (Y.R.); 2020049@hbmzu.edu.cn (J.A.); 1996057@hbmzu.edu.cn (C.T.); 2Hubei Key Laboratory of Selenium Resource Research and Biological Application, Hubei Minzu University, Enshi 445000, China; 2021021@hbmzu.edu.cn; 3Hubei Key Laboratory of Biological Resources Protection and Utilization, Hubei Minzu University, Enshi 445000, China

**Keywords:** air-assisted electrospinning, dextran, zein, dihydromyricetin, Maillard reaction

## Abstract

Dihydromyricetin (DMY) has been encapsulated in delivery systems to address the solubility limitations of DMY in water and improve its bioavailability. Air-assisted electrospinning has been used as a novel technology to load DMY. To evaluate the impact of adding DMY to dextran/zein nanofibers and understand the effects of the Maillard reaction (MR) on the physical and functional properties of DMY-loaded nanofibers, dextran/zein/xylose nanofibers with 0%, 1%, 2%, 3%, and 4% DMY were fabricated, followed by MR crosslinking. Scanning electron microscopy (SEM) observations indicated that the addition of DMY and the MR did not affect the morphology of the nanofibers. X-ray diffraction (XRD) results indicated amorphous dispersion of DMY within the nanofibers and a decreased crystalline structure within the nanofibers following the MR, which might improve their molecular flexibility. The nanofibrous film formed after the MR exhibited both increased tensile strength and elastic modulus due to hydrogen bonding within the nanofibers and increased elongation at break attributed to the increased amorphization of the structure after crosslinking. The nanofibers were also found to exhibit improved heat stability after the MR. The antioxidant activity of the nanofibers indicated a dose-dependent effect of DMY on radical scavenging activity and reducing power. The maintenance of antioxidant activity of the nanofibers after the MR suggested heat stability of DMY during heat treatment. Overall, dextran/zein nanofibers with various DMY contents exhibited tunable physical properties and effective antioxidant activities, indicating that dextran/zein nanofibers offer a successful DMY delivery system, which can be further applied as an active package.

## 1. Introduction

Dihydromyricetin (DMY) is among the main bioactive ingredients in *Ampelopsis grossedentata* leaves, which were regarded as a new food resource by the Chinese government in 2013 [1]. Vine tea is a widely distributed traditional medicinal plant resource in China, exhibiting antioxidant, antibacterial, anti-inflammatory, anticancer, antiobesity, hepatoprotective, and neuroprotective properties [2,3]. To address the solubility limitation of DMY in water and improve its bioavailability, DMY has been encapsulated within nano-capsules [1,4], nano-emulsions [5,6,7], microemulsions [8], hydrogels [9,10,11], and nanofibers [12,13]. Through encapsulation, DMY can be applied as a preservative in food systems. Li et al. [14] developed a chitosan and sodium alginate bilayer film incorporating an inclusion complex of DMY with hydroxypropyl-β-cyclodextrin, which extends the preservation time of fried meatballs by reducing lipid oxidation and microbial growth. Xu et al. [15] prepared a solid dispersion formulation of DMY, thereby extending the shelf life of treated fish stored at 4 °C by more than three days relative to pure DMY. In a previous study, we fabricated gelatin/zein/DMY nanofibers and demonstrated that electrospinning can serve as an effective method to encapsulate DMY [13].

Electrospinning gained greater consideration in nanotechnology due to its flexible and superior processing technique [16]. Air-assisted electrospinning combines electrospinning and solution blowing to facilitate the formation of nanofibers [17] and increases the nanofiber yield [17,18]; it has applications in air filtration [19,20], yarns [21], batteries [22], nanofiber skins [23], and wound healing [24], among others. Along with the ability to increase the yield without compromising quality, it provides a higher degree of control and uniform fiber distribution due to accelerated jet stretching from the airflow. It also promotes rapid solvent evaporation resulting in rapid solidification, significantly enhancing nanofiber quality by producing smooth surface morphologies and excellent fiber mechanical properties. Moreover, it is a practical strategy for scaled-up production, addressing a common limitation faced by conventional electrospinning methods [25,26]. Our previous study used dextran/zein nanofibers obtained by traditional electrospinning for the encapsulation and controlled release of curcumin [27]. Hence, herein, we developed a method to encapsulate DMY within dextran/zein nanofibers via air-assisted electrospinning.

To enhance the performance of biopolymer nanofibers, biopolymer molecules can be modified or combined with additives, such as nanomaterials, crosslinkers, and bioactive compounds, particularly through the Maillard reaction (MR), a promising approach to enhance the performance of packaging for degradable food [28]. The MR is a nonenzymatic browning reaction, which is known to modify the properties of food materials, such as solubility, wettability, and antioxidant activity. This reaction occurs between amino acids and reducing sugars. In previous studies, high temperatures were used for the fabrication of Maillard-crosslinked nanofibers [29,30,31,32,33]; however, elevated temperatures yield conjugates, leading to an irremediable loss of functionality and unregulated proliferation of undesirable MR byproducts. Furthermore, high temperatures may compromise the structural integrity of bioactive elements encapsulated within nanofibers, constricting the application of high-temperature-glycated nanofibers to active packaging. Chen et al. [34] found that DMY decomposed by approximately 33.57% after heating at 100 °C for 720 min. Therefore, contemporary studies have directed their attention toward the exploration of mild MR conditions. In our previous study [35], gelatin/zein/glucose nanofibers were crosslinked at 60 °C and a relative humidity of 50%, resulting in improved water resistance and the formation of stiffer networks. Hence, in this study, we added xylose to dextran/zein solutions as a crosslinker in the MR and explored the effects of the MR on the physical properties and antioxidant activity of DMY-loaded dextran/zein nanofibers. In essence, this integration of air-assisted electrospinning with a modulated Maillard reaction introduces a novel approach to encapsulating dihydromyricetin within nanofibers. This encapsulation broadens the spectrum of practical applications while maintaining and even enhancing the antioxidant properties essential in several domains.

Thus, we fabricated dextran/zein/xylose nanofibers encapsulating 0–4% DMY, followed by the MR at mild conditions. The prepared nanofibers were characterized by scanning electron microscopy (SEM), Fourier transform infrared (FTIR) spectroscopic analysis, thermal analysis, X-ray diffraction (XRD), and water contact angle measurements. The mechanical properties and water vapor permeability (WVP) were tested to demonstrate the potential of DMY as a food package. The antioxidant activities of the nanofibers were assessed by the 2,2-diphenyl-1-picrylhydrazyl (DPPH) method, the 2,2′-azino-bis (3-ethylbenzothiazoline-6-sulfonic acid) (ABTS) method, the ferric reducing antioxidant power (FRAP) assay, and the cupric reducing antioxidant capacity (CUPRAC) assay.

## 2. Results and Discussion

### 2.1. Fiber Morphologies

Figure 1 shows the morphology of the dextran/zein nanofibers without or with various DMY contents. The diameter histograms are shown in Appendix A. When the DMY encapsulation ratio was increased from 0% to 4%, the nanofiber diameter increased slightly from 735.8 to 782.4 nm; however, no other significant differences were observed, indicating that DMY did not affect the morphology of the nanofibers. Liu et al. [12] also observed a slightly increased diameter of PVP/chitosan nanofibers after the incorporation of DMY. Fibers crosslinked at 60 °C and 50% relative humidity for 6 h retained their morphologies. Heat treatment under these conditions did not destroy the morphology or affect the porosity of the nanofibers, in accordance with our previous results [35]. The maintenance of the nanofiber morphology and surface-to-volume ratio after crosslinking facilitated the application of the nanofibers as effective encapsulation systems for bioactive applications.

### 2.2. FTIR Spectroscopy

The FTIR results of the nanofibers before and after the MR are shown in Figure 2. The peaks at approximately 3200–3300 cm^−1^ reflect characteristic absorptions of hydroxyl groups. Peaks at 2800–2950 cm^−1^ correspond to C–H asymmetric stretching. A slight shift of a peak at 2887 to 2923 cm^−1^ was observed after the incorporation of DMY before the MR. Amides I and II are indicated by peaks centered at 1652 and 1539 cm^−1^, respectively. The band corresponding to C–O stretching shifted from 1011 cm^−1^ for DX0 to 1015 cm^−1^ for nanofibers with DMY, which might be attributed to the formation of hydrogen bonds between the hydroxyl groups on the outside of the dextran and the phenolic hydroxyl groups of DMY.

### 2.3. XRD Analysis

Figure 3 shows the diffraction pattern of DMY-loaded nanofibers before (a) and after (b) the MR. The XRD patterns of the nanofibers before crosslinking were observed to feature one broad peak of approximately 20°, indicating a typical non-crystalline structure. It has been reported that DMY exhibits a characteristic diffraction peak at 16.7° [34]; the absence of this diffraction peak indicates amorphous dispersion of DMY within nanofibers. Previous studies also supported that the electrospinning process facilitated the formation of an amorphous structure. Figure 3b shows two relatively narrow peaks at 10° and 18°, the intensities of which decreased significantly after crosslinking. Studies have shown that substances with low crystallinity tend to exhibit higher solubility and bioavailability [36]. The decreased crystalline structure within nanofibers facilitates solubility and bioavailability, thereby broadening their scope of application. Dextran molecules contain many hydroxyl groups, which may interact with the phenolic hydroxyl groups on DMY to form hydrogen bonding. Similar suppression of the formation of crystals from bioactive materials within electrospinning nanofibers has also been reported [27,34,37]. Zhang et al. [38] observed a decreased diffraction peak of gluten/zein nanofibers after the MR.

### 2.4. Thermal Analysis

Figure 4 and Figure 5 show the DSC, TGA, and derivation of TGA curves of the nanofibers before and after the MR, respectively. Detailed data are summarized in Table 1. The DSC curves show a broad endothermic peak, which is related to denaturation of the nanofibers. The denaturation temperature before crosslinking increased from 80.6 °C in DX0 to 83.1 °C, 86.3 °C, and 83.3 °C in DX2, DX3, and DX4, respectively, indicating improved heat stability of the nanofibers with DMY. Interactions between dextran and DMY by hydrogen bonding may explain the increased denaturation temperature. After the MR, the nanofibers exhibited decreased denaturation temperatures and enthalpies, which can be explained by the fact that the dextran/zein nanofibers had undergone denaturation upon MR heating. Table 1 shows that heat degradation of the nanofibers primarily occurred in three stages. The primary phase can be largely ascribed to the expulsion of bound water from the nanofibers. The second peak can be ascribed to decomposition of the proteins [37]. The third peak corresponded to the main thermal degradation of dextran and zein. It can also be observed that, after the MR, the nanofibers showed higher degradation temperatures, indicating increased thermal stability after Maillard crosslinking. Similar improvements in heat stability were observed after the MR by others using gluten/zein [38] and gelatin/zein nanofibers [35].

### 2.5. Water Contact Angle Analysis

The water contact angles of the nanofibers were measured at the water drop contact of the film (Figure 6). With the addition of DMY, the water contact angle increased at 1% DMY content and then decreased above this value. Wang et al. [39] observed a similar decrease in water contact angles after the incorporation of 2–10% DMY in PCL nanofibers. A similarly decreased water contact angle was observed in PVP/chitosan/DMY nanofibers [12].

### 2.6. Mechanical Properties

Figure 7 shows the mechanical properties of the nanofibrous films in terms of tensile strength, elongation at break, and elastic modulus. Figure 7a shows that the tensile strength of the DMY-loaded nanofibrous films increased significantly following the MR. No significant differences were observed among nanofibers with various DMY contents before the MR. DX2M6 and DX3M6 showed higher tensile strengths compared to those of the uncrosslinked nanofibers and those with other DMY contents. Elongation at break reflects the flexibility of molecules within nanofibers. With the addition of DMY, the elongation at break decreased significantly from 22.65% for DX0 to 6.63%, 4.43%, 5.5%, and 6.23% for DX1, DX2, DX3, and DX4, respectively. The MR not only improved the strength of the dextran/zein nanofibers but also the flexibility of those with 2% and 3% DMY contents. The elastic modulus increased slightly after the MR, which is ascribed to intermolecular entanglements and interactions among molecules due to hydrogen bonding [40]. As indicated by the XRD results, a more amorphous structure was formed after the MR, which facilitated the slippage of molecules within nanofibers upon tensile testing. Crosslinked nanofibers with 2% DMY content exhibited the highest elastic modulus of 40.88 MPa. Xie et al. [41] observed that DMY decreased the elongation at break of konjac glucomannan/gellan gum films, and that the elastic modulus increased after 3% DMY addition. Wang et al. [39] found that the addition of DMY in PCL nanofibers at concentrations of 2%, 6%, and 10% increased the elastic modulus and decreased the elongation at break.

### 2.7. Water Vapor Permeability

Figure 8 shows the WVP of the dextran/zein nanofibers before and after the MR. The incorporation of DMY at contents of 1% and 2% decreased the WVP slightly; however, the WVP increased at concentrations of 3% and 4%. The increased WVP of DX4 may be related to increased hydrophobicity, as indicated by the water contact angle results. The MR led to a significant increase in WVP, which can be explained by the increased water stability after the MR such that the porous structure is maintained upon water vapor diffusion [35]. Packages with high WVP are suitable for use pertaining to fresh foods, while those with low WVP meet the requirements for processed food packaging [42]. Moreover, high WVP is favored for wound exudates. Hence, crosslinked DMY-loaded dextran/zein nanofibers might be suitable for use in fresh food packaging or wound-healing applications.

### 2.8. Antioxidant Activities

Antioxidant activities were assessed in terms of radical scavenging activity (RSA) and reducing power (Figure 9). Nanofibers with various contents of DMY showed a DMY dose-dependent effect on DPPH RSA. Compared to the uncrosslinked nanofibers, those with DMY exhibited significantly higher DPPH RSA after the MR, which can be explained by the amorphous structure facilitating the function of DMY. However, nanofibers without DMY also exhibited higher DPPH and ABTS RSA, which was ascribed to the formation of MR products with antioxidant activities [43,44]. The ferric and cupric reducing powers of the nanofibers also exhibited a dose effect, and no significant decrease was observed after the MR. The decomposition of dihydromyricetin has been reported to result in decreased antioxidant activity and bioactivity [34,45]. Vice versa, the maintenance of RSA and reducing power after the MR indicated that the dextran/zein nanofibers can serve as good encapsulation systems for DMY, protecting it from decomposing upon heating.

## 3. Materials and Methods

### 3.1. Chemicals

Dextran (MW ~70 kDa) was purchased from Macklin Biochemical Technology Co., Ltd. (Shanghai, China). Zein (grade Z3625, 22–24 kDa), TPTZ (2,4,6-tri(2-pyridinyl)-1,3,5-triazine, purity ≥ 98%), DPPH (2,2-diphenyl-1-picrylhydrazyl, purity ≥ 98%), neocuproine (2,9-dimethyl-1,10-phenanthroline, purity ≥ 98%), and ABTS (2,2′-azino-bis (3-ethylbenzothiazoline-6-sulfonic acid, purity ≥ 98%) were purchased from Sigma Aldrich (St. Louis, MO, USA). Dihydromyricetin (purity > 97%), xylose (purity ≥ 99%), and acetic acid (purity 99.8%) were purchased from Aladdin Reagent Database Inc. (Shanghai, China).

### 3.2. Solution Preparation

Dextran (50% *w*/*v*) was dissolved in a 50% acetic acid aqueous solution with 5% (*w*/*v*) zein. Xylose (5% *w*/*v*) was then mixed with this dextran/zein solution. DMY was added at concentrations of 1%, 2%, 3%, and 4% with respect to the combined weight of dextran and zein. All solutions were stirred overnight to ensure complete dissolution.

### 3.3. Air-Assisted Electrospinning

A 10 mL syringe was used as the solution container, and the solution was continuously pumped into a 20 G needle tip at a speed of 10 mL/h under the action of a pressure pump (Figure 10). The solution was stretched into fibers under the effect of airflow with a flow rate of 400 L/h and an electrostatic field of 18 kV. The fibers were then collected at a distance of 20 cm from the needle tip.

According to the amount of DMY added, the prepared samples with 0, 1%, 2%, 3%, and 4% DMY were denoted as DX0, DX1, DX2, DX3, and DX4, respectively.

### 3.4. Maillard Reaction

Nanofibrous films were crosslinked by MR conducted in a temperature- and humidity-controlled chamber (HSX-150L, Shanghai Gipp Electronic Technology Co., Ltd., Shanghai, China) operating at a temperature of 60 °C and relative humidity (RH) of 50% for 6 h.

### 3.5. Fiber Morphologies

After vacuum gold-spraying treatment, the morphology of the fibers was observed via SEM. Using Nano Measure 1.2 software, fibers were randomly selected from SEM images, and their diameter distribution was determined.

### 3.6. FTIR Spectroscopy

FTIR spectra were acquired using an FTIR spectrometer (iS5, Thermo Nicolet Ltd., Waltham, MA, USA) operating in ATR mode over the wavenumber range of 600–4000 cm^−1^ with a resolution of 2 cm^−1^; 32 FTIR scans were performed for each analysis.

### 3.7. XRD Analysis

The crystallization behavior of the nanofibers was investigated via XRD (Rigaku Smart Lab SE diffractometer, Rigaku Smart Lab, Rigaku Corporation, Tokyo, Japan) using Cu K-α radiation over a 2θ range of 5–90° (at a rate of 1°/min).

### 3.8. Analysis of Thermal Properties

The thermal properties of the nanofibers were analyzed using a TG–DSC analyzer (NETZSCH-Gerätebau GmbH, Selb, Germany). For this purpose, 6–10 mg of the samples was accurately weighed and placed in a crucible. Further, an empty crucible maintained under the same conditions was used as a reference.

### 3.9. Water Contact Angle Tests

A tensiometer (OCA 20, Dataphysics Instruments, Filderstadt, Germany) was used to test the water contact angle of the nanofibers. A syringe was used to deposit 3 μL of droplets onto the surface of the nanofibers (20 × 40 × 0.2 mm). The water contact angle was measured after the droplets came into contact with the surface of the nanofibers. The measurements were repeated three times for each sample.

### 3.10. Mechanical Properties

The thickness, width, and intercept of the dextran/zein nanofibrous films were tested by a digital micrometer. Then, the tensile strength (TS), elastic modulus (EM), and elongation at break (EB) were measured by a DR-508A (Dongri Instrument Ltd., Dongguan, China) computer tensile testing machine with a tensile load of 5 N and a tensile rate of 5 mm/min [46]. The tests were conducted five times for each sample, and the tensile strength, elongation at break, elastic modulus were calculated as follows:$$\mathrm{Tensile}\mathrm{strength}\;\left(\mathrm{MPa}\right)=\frac{\mathrm{Load}\;\mathrm{at}\;\mathrm{Break}}{\mathrm{Original}\;\mathrm{width}\;\times \mathrm{Original}\;\mathrm{thickness}}\;\mathrm{Elongation}\;\mathrm{at}\;\mathrm{break}\;\left(\%\right)=\frac{\mathrm{Elongation}\;\mathrm{at}\;\mathrm{rupture}}{\mathrm{Original}\;\mathrm{test}\;\mathrm{length}}\;\times 100\;\mathrm{Elastic}\;\mathrm{modulus}\;\left(\mathrm{MPa}\right)=\frac{\mathrm{Stress}}{\mathrm{Strain}}$$

### 3.11. Water Vapor Permeability

WVP was assessed utilizing the ASTM E96 gravimetric technique, following a modified version of the amended method proposed by Zhang et al. [47]. The rim of a permeable cup with a capacity of 10 mL was secured with a consistently thick, 6 cm diameter nanofiber film, which was then placed in a desiccator. For all samples, weights were logged on an hourly basis. This procedure was conducted thrice over the span of 6 h. The tests were repeated three times. The calculation for WVP was as follows [48]:$$\mathrm{WVP}\left(g/m\cdot s\cdot \mathrm{pa}\right)=\frac{W_{s}}{\mathrm{At}\;}\times \frac{L}{∆P}$$

Here, A represents the contact area (measured in cm^2^); L is the thickness of the nanofibrous film (in cm); ∆P refers to the rated vapor pressure differential (in Pa) with a set value of 2237.8 Pa at 28 °C; and W_s_/t indicates the linear regression of weight over time (measured in g/s).

### 3.12. Antioxidant Activity

Antioxidant activities were measured by DPPH, ABTS, FRAP, and CUPRAC assays according to previous methods [13]. All the measurements were conducted three times. Statistical analysis of data was carried out using Origin 8.0 software (OriginLab, Northampton, MA, USA), and one-way analysis of variance (ANOVA) using the Tukey’s test with *p* values < 0.05 considered significant was performed for comparative analysis.

#### 3.12.1. DPPH Assay

A DPPH solution, with a concentration of 500 μmol/L, was prepared immediately before use by diluting 0.0195 g of DPPH in 100 mL of ethanol. A sample weighing 5 mg was mixed with 2 mL of the DPPH solution and allowed to react under dark conditions for 30 min. Following this, A517 nm was measured and used to calculate the scavenging rate as follows:$$\mathrm{DPPH}\;\mathrm{radical}\;\mathrm{scavenging}\;\mathrm{rate}\;\left(\%\right)=\frac{A_{0}-A_{1}}{A_{0}}\times 100$$
where A_0_ is A517 nm of the blank solution and A_1_ is A517 nm of the sample solution.

#### 3.12.2. ABTS Assay

To prepare the ABTS and K_2_S_2_O_8_ solutions, 0.0384 g of ABTS and 0.0134 g of K_2_S_2_O_8_ were separately dissolved in 10 mL of deionized water. These solutions were combined in a 1:1 ratio and allowed to react under dark conditions for 12 h to yield the active ABTS solution. This solution was then further diluted using a pH 7.4 PBS solution until the resulting ABTS/PBS solution achieved an absorbance of 0.7 ± 0.02 at 734 nm. A 10 mL centrifuge tube was filled with 5 mg of the sample, and 2 mL of the ABTS/PBS solution was added to the tube. The solution was left to react under dark conditions for 30 min, after which A734 nm was measured. From these results, the rate of free radical scavenging was determined as follows:$$\mathrm{ABTS}\;\mathrm{radical}\;\mathrm{scavenging}\;\mathrm{rate}\;\left(\%\right)=\frac{A_{0}-A_{1}}{A_{0}}\times 100$$
where A_0_ is A734 nm of the blank solution and A_1_ is A734 nm of the sample solution.

#### 3.12.3. FRAP Assay

Three aqueous solutions were initially prepared: HCl solution at a concentration of 40 mmol/L; FeCl_3_ solution with a concentration of 20 mmol/L; and 0.3 mol/L CH3COONa buffer adjusted to pH 3.6. Subsequently, 10 mL of the HCl solution (40 mmol/L) was used to dilute 0.03123 g of 2,4,6-Tri(2-pyridyl)-1,3,5-triazine (TPTZ) to create a TPTZ derivative solution. Using a 10:1:1 ratio, the CH_3_COONa buffer, FeCl3 solution, and TPTZ solution were mixed to obtain the active TPTZ solution. A 5 mg sample was weighed into a 10 mL centrifuge tube, supplemented with 2 mL of the TPTZ working solution, and allowed to undergo a reaction at 37 °C for a duration of 30 min. Parallel absorbance measurements were recorded at 592 nm. The ferric reducing power is denoted by the A592 nm values.

#### 3.12.4. CUPRAC Assay

The 0.01 mol/L CuCl_2_ aqueous solution, 1 mol/L CH_3_COONH_4_ aqueous solution, and 7.5 mmol/L C_14_H_12_N_2_ ethanol solution were prepared and mixed at a ratio of 1:1:1 to prepare the working solution. A total of 5 mg of the sample was weighed and mixed with 2 mL of the working solution at 25 °C for 30 min, and A450 nm was measured. The cupric reducing ability is reflected by the A450 nm value.

## 4. Conclusions

In this work, DMY-loaded dextran/zein nanofibers were fabricated by air-assisted electrospinning. The MR conducted at 60 °C with an RH of 50% was shown to be an adequate condition for crosslinking dextran/zein/DMY nanofibers such that DMY would not decompose upon heating. The antioxidant activity of the nanofibers showed a dose-dependent effect of DMY on radical scavenging activity and reducing power. The maintenance of the antioxidant activity of the nanofibers after the Maillard reaction suggests the heat stability of DMY during heat treatment. The amorphous dispersion of DMY within the nanofibers and the low crystalline structure of the nanofibers would solve the low solubility limitation and facilitate the function and application of dihydromyricetin as a food package or wound dressing. Future research is recommended to delve further into their stability profiles in more diverse conditions to provide a comprehensive understanding of the nanofibers.

## Figures and Tables

**Figure 1 molecules-29-03136-f001:**
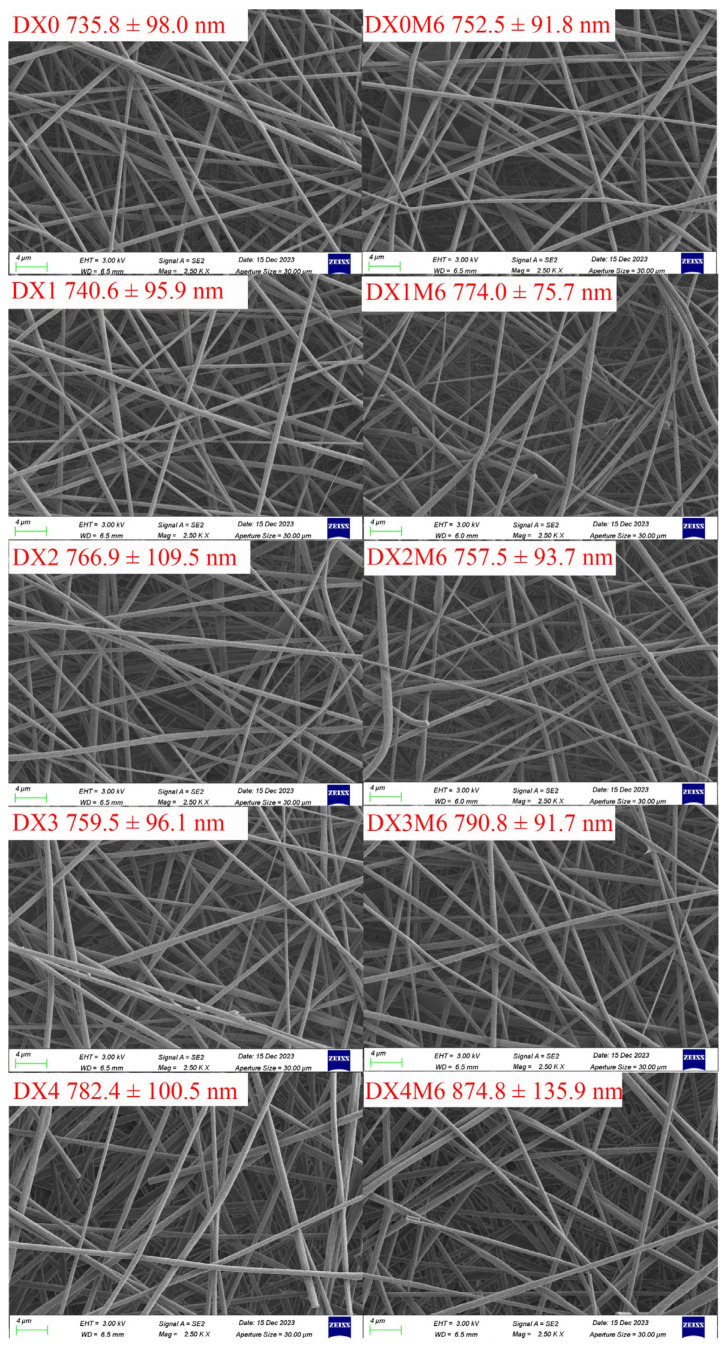
Morphologies of dextran/zein nanofibers without (DX0) DMY and with 1% (DX1), 2% (DX2), 3% (DX3), and 4% (DX4) DMY. Nanofibers crosslinked for 6 h are denoted as DX0M6, DX1M6, DX2M6, DX3M6, and DX4M6.

**Figure 2 molecules-29-03136-f002:**
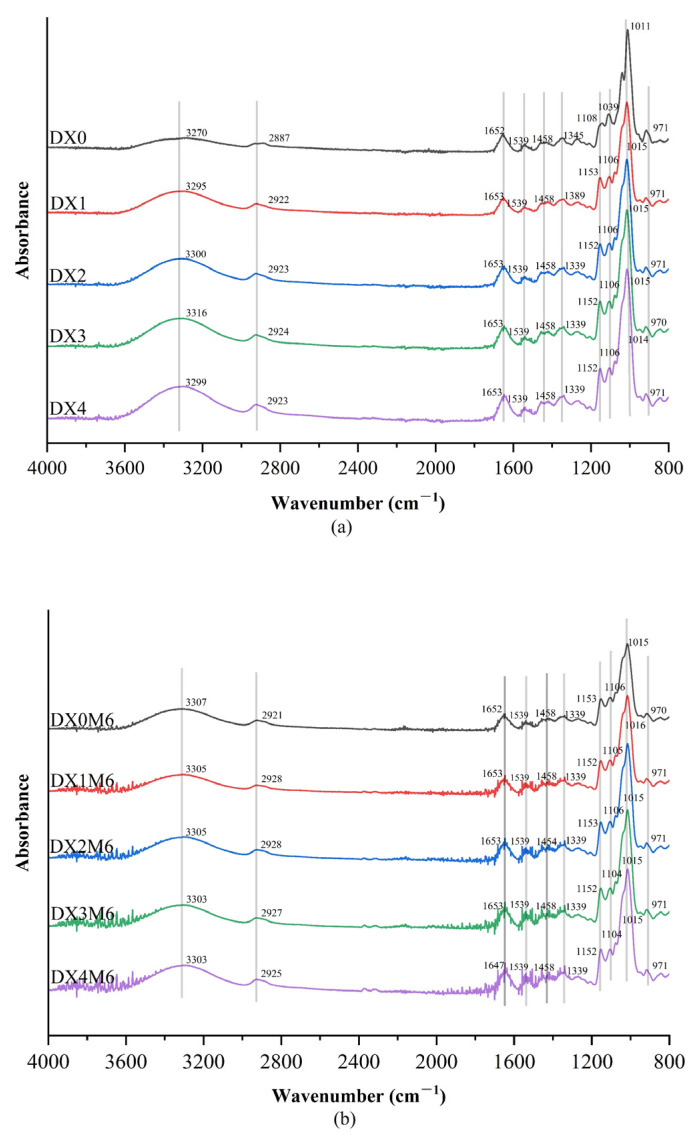
FTIR spectra of dextran/zein nanofibers without (DX0) DMY and with 1% (DX1), 2% (DX2), 3% (DX3), and 4% (DX4) DMY before (**a**) and after (**b**) crosslinking.

**Figure 3 molecules-29-03136-f003:**
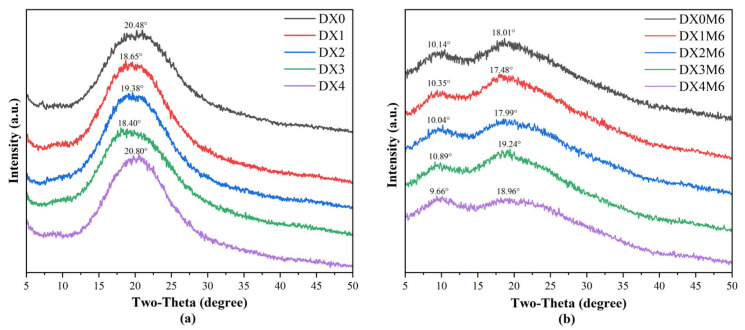
XRD patterns of dextran/zein nanofibers without (DX0) DMY and with 1% (DX1), 2% (DX2), 3% (DX3), and 4% (DX4) DMY before (**a**) and after (**b**) crosslinking.

**Figure 4 molecules-29-03136-f004:**
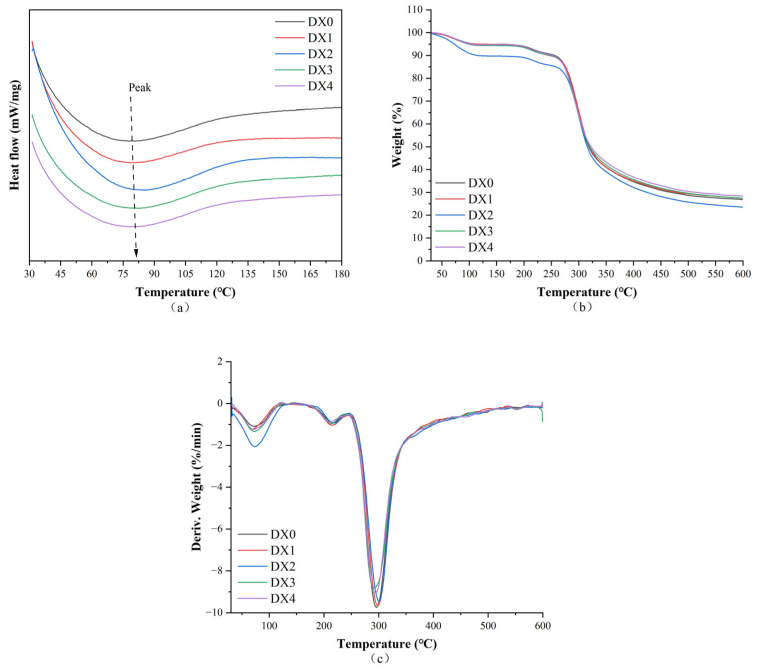
Thermal curves of dextran/zein nanofibers without (DX0) DMY and with 1% (DX1), 2% (DX2), 3% (DX3), and 4% (DX4) DMY. (**a**) DSC; (**b**) TGA; and (**c**) DTG curves.

**Figure 5 molecules-29-03136-f005:**
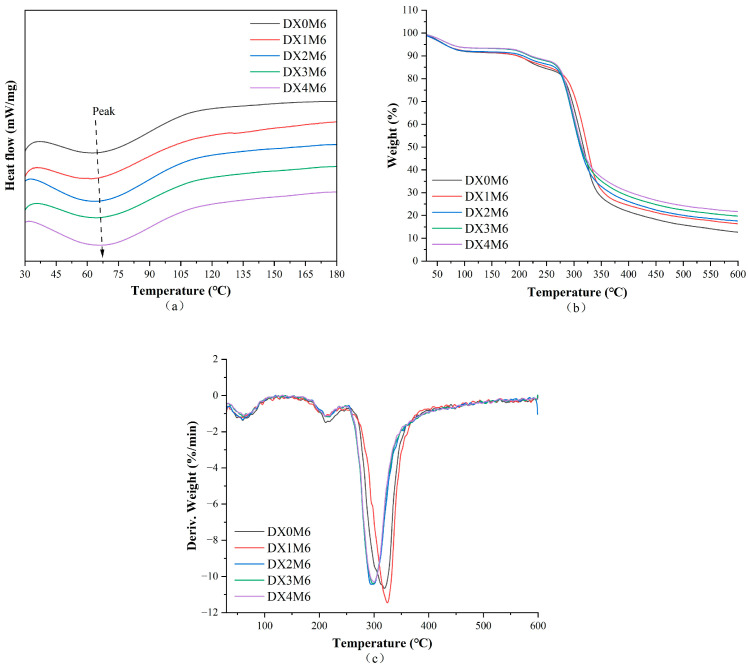
Thermal curves of dextran/zein nanofibers without (DX0M6) DMY and with 1% (DX1M6), 2% (DX2M6), 3% (DX3M6), and 4% (DX4M6) DMY after crosslinking for 6 h. (**a**) DSC; (**b**) TGA; and (**c**) DTG curves.

**Figure 6 molecules-29-03136-f006:**
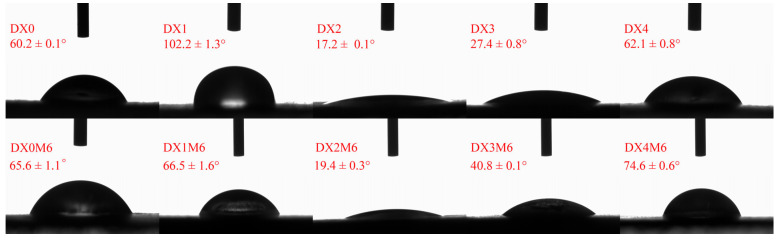
Water contact angles of dextran/zein nanofibers without (DX0) DMY and with 1% (DX1), 2% (DX2), 3% (DX3), and 4% (DX4) DMY and those of the nanofibers crosslinked at 60 °C and 50% RH for 6 h.

**Figure 7 molecules-29-03136-f007:**
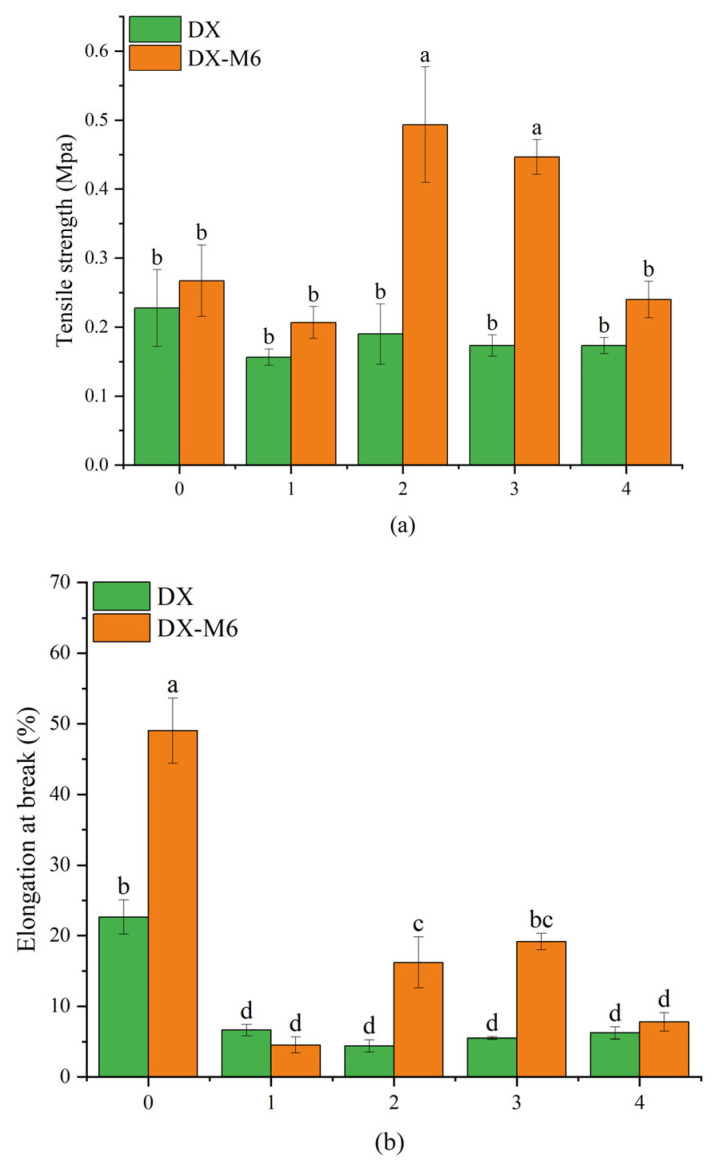
Tensile test results of dextran/zein nanofibers without (DX0) DMY and with 1% (DX1), 2% (DX2), 3% (DX3), and 4% (DX4) DMY, as well as of nanofibers crosslinked for 6 h. (**a**) Tensile strength; (**b**) elongation at break; and (**c**) elastic modulus. Different letters indicate significant difference (*p* < 0.05) between samples.

**Figure 8 molecules-29-03136-f008:**
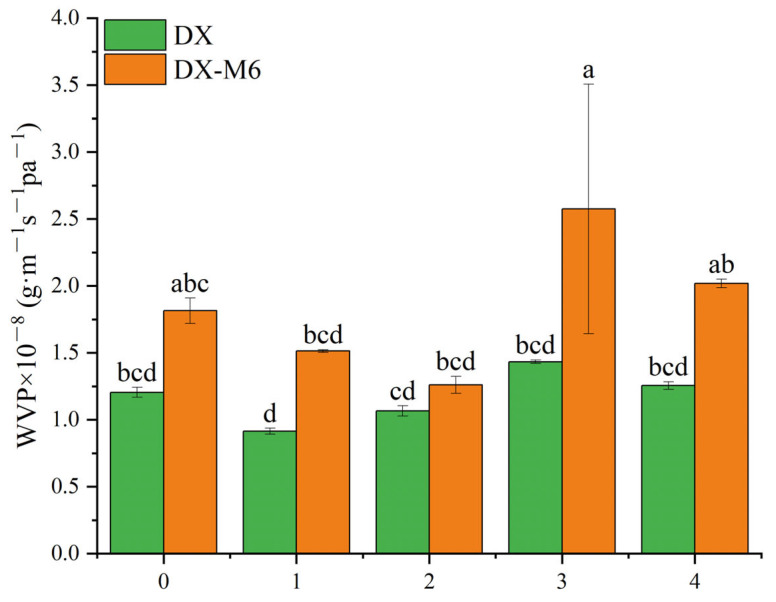
WVP of dextran/zein nanofibers without DMY (DX0) and with 1% (DX1), 2% (DX2), 3% (DX3), and 4% (DX4) DMY, as well as of the nanofibers crosslinked for 6 h. Different letters indicate significant difference (*p* < 0.05) between samples.

**Figure 9 molecules-29-03136-f009:**
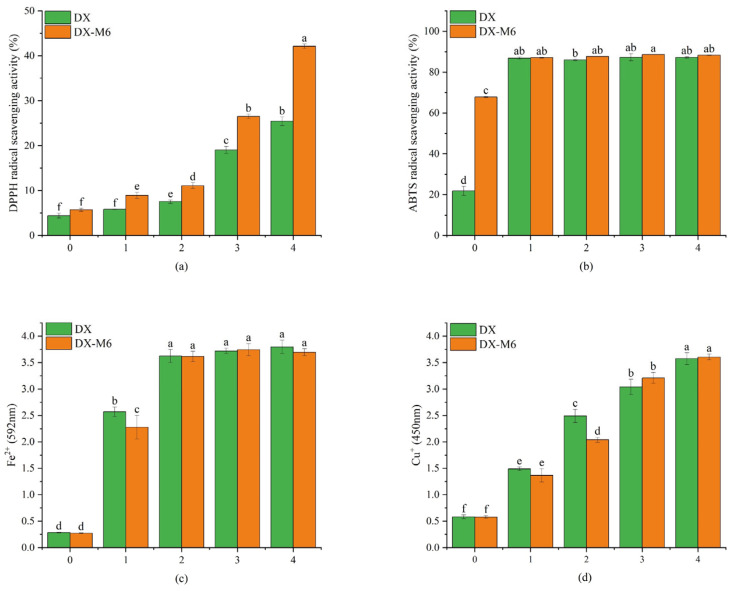
Antioxidant activities of the dextran/zein nanofibers without DMY (DX0) or with 1% (DX1), 2% (DX2), 3% (DX3), and 4% (DX4) DMY, as well as of the nanofibers crosslinked for 6 h. (**a**) DPPH RSA; (**b**) ABTS RSA; (**c**) ferric reducing activity; and (**d**) copper reducing activity. Different letters indicate significant difference (*p* < 0.05) between samples.

**Figure 10 molecules-29-03136-f010:**
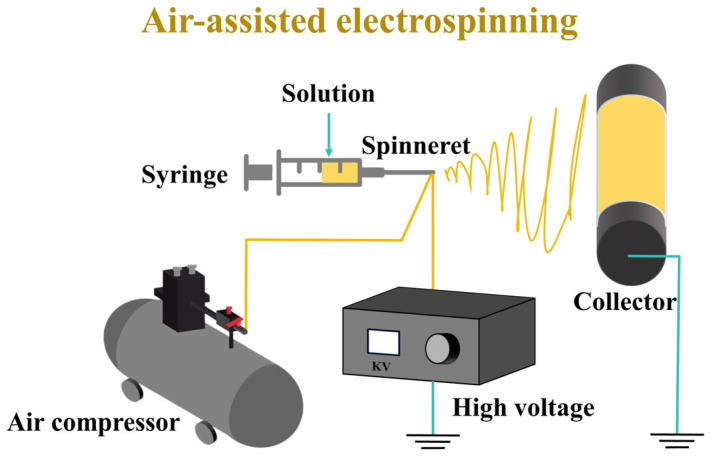
Schematic illustration of the air-assisted electrospinning setup.

**Table 1 molecules-29-03136-t001:** DSC and TGA data of dextran/zein nanofibers without (DX0) DMY and with 1% (DX1), 2% (DX2), 3% (DX3), and 4% (DX4) DMY and those of the nanofibers after crosslinking for 6 h.

	DSC		TGA						
	T (°C)	∆H (J/g)	Peak 1 (°C)	Weight Loss (%)	Peak 2 (°C)	Weight Loss (%)	Peak 3 (°C)	Weight Loss (%)	Residue at 600 °C (%)
DX0	80.6	−29.086	73.3	5.14	216.0	4.61	296.1	63.39	26.86
DX1	80.8	−25.474	69.4	5.11	214.7	6.50	297.9	60.97	27.42
DX2	83.1	−31.134	70.9	10.24	215.1	5.53	297.6	60.77	23.45
DX3	86.3	−28.642	72.8	5.63	213.2	4.66	290.9	62.25	27.42
DX4	83.3	−28.798	72.4	5.28	210.1	5.25	294.3	61.08	28.39
DX0M6	69.4	−6.303	59.8	8.55	211.7	7.35	319.1	71.41	12.69
DX1M6	67.1	−4.748	53.9	8.35	210.6	6.53	324.4	68.74	16.38
DX2M6	67.9	−8.643	59.8	8.18	218.2	5.66	297.6	68.60	17.56
DX3M6	67.5	−7.732	60.9	6.70	217.6	5.50	298.6	68.06	19.74
DX4M6	70.2	−8.491	63.4	6.59	210.8	5.36	300.2	66.34	21.71

## Data Availability

Research data are not shared.

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
