# Peer review of "Air-Assisted Electrospinning of Dihydromyricetin-Loaded Dextran/Zein/Xylose Nanofibers and Effects of the Maillard Reaction on Fiber Properties"

_molecules, 2024, doi:10.3390/molecules29133136_

Round 1
Reviewer 1 Report
Comments and Suggestions for Authors
Some minor revisions are necessary before publication:
1. There are some typographical and grammatical errors that should not be avoided.
2. In Figure 1. Error bar should be presented.
3. The rational design of electrospun nanofiber mat should be explain in the introduction section with the reference of J. Compos. Sci. 2024, 8(4), 127; https://doi.org/10.3390/jcs8040127
4. References should be given to formula equations.
5. Purity of chemicals should be mentioned in the revised manuscript.
Comments on the Quality of English LanguageThere are some typographical and grammatical errors that should not be avoided.
Author Response
Some minor revisions are necessary before publication:
- There are some typographical and grammatical errors that should not be avoided.
Response: Thanks for your comments. We have asked professionals to revise the English of the manuscript.
- In Figure 1. Error bar should be presented.
Response: The histogram figure is the diameter distribution histogram, which has no error bar. But the histogram at here is not very clear, so we separated it and put it in the supplementary materials.
- The rational design of electrospun nanofiber mat should be explain in the introduction section with the reference of J. Compos. Sci. 2024, 8(4), 127; https://doi.org/10.3390/jcs8040127
Response: We have added the reference to support our point.
- References should be given to formula equations.
Response: We have checked the formula equations in the manuscript and added the references accordingly.
- Purity of chemicals should be mentioned in the revised manuscript.
Response: Thanks for the comments. We have added the purity of the chemicals in materials.
Reviewer 2 Report
Comments and Suggestions for Authors
The aim of the manuscript entitled "Air-assisted electrospinning of dihydromyricetin encapsulated dextran/zein nanofiber and effect of Maillard reaction on fiber properties" is to explore the application of dihydromyricetin-loaded nanofibers for potential use in food packaging or wound dressing. Utilizing air-assisted electrospinning, the authors effectively encapsulated dihydromyricetin into dextran/zein nanofibers. Maillard reaction under specific conditions was employed for crosslinking these nanofibers, preventing the decomposition of dihydromyricetin upon heating. The results demonstrate that the crosslinked nanofibers retained their excellent antioxidant activity, indicating the potential use of these materials for protecting dihydromyricetin from thermal degradation. The amorphous dispersion of dihydromyricetin within the nanofibers and the low crystalline structure of the nanofibers further enable their application in addressing issues of low solubility and enhancing functionality. However, further detailed analyses may be beneficial to confirm the stability and efficacy of these materials in their practical applications.
The manuscript is interesting and fits well with the scope of the Journal. The manuscript is generally well-prepared. My specific comments are given below.
The introduction is informative, but the novelty of this work should be written at the end of the introduction. The conclusion should be rewritten to include a summary of the study's findings, their interpretation and significance, limitations, and suggestions for future research. Since the study presents promising results regarding the encapsulation of dihydromyricetin in nanofibers and its potential applications, some aspects could be improved, for example, investigating stability profiles to assess the long-term stability of the nanofibers under various environmental conditions, including temperature and humidity variations.
Author Response
The aim of the manuscript entitled "Air-assisted electrospinning of dihydromyricetin encapsulated dextran/zein nanofiber and effect of Maillard reaction on fiber properties" is to explore the application of dihydromyricetin-loaded nanofibers for potential use in food packaging or wound dressing. Utilizing air-assisted electrospinning, the authors effectively encapsulated dihydromyricetin into dextran/zein nanofibers. Maillard reaction under specific conditions was employed for crosslinking these nanofibers, preventing the decomposition of dihydromyricetin upon heating. The results demonstrate that the crosslinked nanofibers retained their excellent antioxidant activity, indicating the potential use of these materials for protecting dihydromyricetin from thermal degradation. The amorphous dispersion of dihydromyricetin within the nanofibers and the low crystalline structure of the nanofibers further enable their application in addressing issues of low solubility and enhancing functionality. However, further detailed analyses may be beneficial to confirm the stability and efficacy of these materials in their practical applications.
The manuscript is interesting and fits well with the scope of the Journal. The manuscript is generally well-prepared. My specific comments are given below.
The introduction is informative, but the novelty of this work should be written at the end of the introduction. The conclusion should be rewritten to include a summary of the study's findings, their interpretation and significance, limitations, and suggestions for future research. Since the study presents promising results regarding the encapsulation of dihydromyricetin in nanofibers and its potential applications, some aspects could be improved, for example, investigating stability profiles to assess the long-term stability of the nanofibers under various environmental conditions, including temperature and humidity variations.
Response: Thanks for the comments. We have added a part in the introduction to empathize the novelty of this work. And the conclusion part has been written as the reviewers’ comments.
Reviewer 3 Report
Comments and Suggestions for Authors
Manuscript ID: molecules-3056822
Air-assisted electrospinning of dihydromyricetin encapsulated dextran/zein nanofiber and effect of Maillard reaction on fiber properties
Yupeng Ren, Jianhui An, Cheng Tian, Longchen Shang, Yexing Tao, Lingli Deng
Comments and Suggestions for Authors
To develop biodegradable and environment-friendly functional food-packaging materials, gelatin/zein/xylose nanofibers were fabricated through air-assisted electrospinning and cross-linked by the Maillard reaction in this study. The manuscript contains relevant information that may be of interest to readers and is suitable for publication. However, it needs to be revised before publication. Here are some important comments:
1. The title of the manuscript should reflect the addition of xylose to dextran/zein solutions as a cross-linker in the Maillard reaction. The authors should also reconsider the word "xylose" in the title and text as well, following Liu, S.; Luo S.; Li Y.; Zhang H.; Yuan Z.; Shang L., Deng L. Influence of the Maillard Reaction on Properties of Air-Assisted Electrospun Gelatin/Zein/Glucose Nanofibers. Foods 2023. 12(3): 45.
2. Why is Glucose mentioned in chemicals?
3. All figure titles are misspelled. Please revise them and include the probable w% percentages of dihydromyricetin.
4. Please include a schematic illustration of the air-assisted electrospinning process in the section about the electrospinning process.
5. Where possible, please add the number of determinations for each sample and use a statistical evaluation method in the relevant section.
6. Provide discussions on the weight loss that occurred in three steps as shown in Figures 4 and 5. Are there any other endothermic reactions occurring at temperatures higher than 175 °C in the DSC results?
7. Standardize the degree sign as ° throughout the manuscript and correct: the electrostatic field adjusted to 18 kV. Additionally, correct other spelling and grammatical mistakes.
Comments on the Quality of English Language
Minor editing of English language required.
Author Response
To develop biodegradable and environment-friendly functional food-packaging materials, gelatin/zein/xylose nanofibers were fabricated through air-assisted electrospinning and cross-linked by the Maillard reaction in this study. The manuscript contains relevant information that may be of interest to readers and is suitable for publication. However, it needs to be revised before publication. Here are some important comments:
- The title of the manuscript should reflect the addition of xylose to dextran/zein solutions as a cross-linker in the Maillard reaction. The authors should also reconsider the word "xylose" in the title and text as well, following Liu, S.; Luo S.; Li Y.; Zhang H.; Yuan Z.; Shang L., Deng L. Influence of the Maillard Reaction on Properties of Air-Assisted Electrospun Gelatin/Zein/Glucose Nanofibers. Foods 2023. 12(3): 45.
Response: We have revised the title as your suggestion.
- Why is Glucose mentioned in chemicals?
Response: Sorry for the mistake, glucose was not used in this study.
- All figure titles are misspelled. Please revise them and include the probable w% percentages of dihydromyricetin.
Response: Thanks for your comments. We believe the figure captions are clear. We have mentioned the percentages of dihydromyricetin in figure captions. The dihydromyricetin was add at the ratio of 1%, 2%, 3%, and 4%, which respect to the whole weight of dextran and zein. According to the amount of dihydromyricetin added, the prepared samples with 0, 1%, 2%, 3%, and 4% dihydromyricetin were denoted DX0, DX1, DX2, DX3, and DX4, respectively.
- Please include a schematic illustration of the air-assisted electrospinning process in the section about the electrospinning process.
Response: We have added a schematic illustration of the air-assisted electrospinning process in the materials and methods section.
- Where possible, please add the number of determinations for each sample and use a statistical evaluation method in the relevant section.
Response: We have added the number of determinations for each sample. Statistical analysis evaluation method was added in section 3.12.
- Provide discussions on the weight loss that occurred in three steps as shown in Figures 4 and 5. Are there any other endothermic reactions occurring at temperatures higher than 175 °C in the DSC results?
Response: We have analyzed the three steps of TG curve and revised the data in Table 1. We have added the discussion about it.
There is no endothermic peak observed above 175 °C, while the curves of DSC decreased constantly with the temperature increased, indicating the occurrence of endothermic reactions, corresponding to the decomposition of the nanofibers.
- Standardize the degree sign as ° throughout the manuscript and correct: the electrostatic field adjusted to 18 kV. Additionally, correct other spelling and grammatical mistakes.
Response: Thanks for your comments. We have carefully checked the whole manuscript and revised it accordingly.
Minor editing of English language required.
Response: Thanks for your comments. We have asked professionals to revise the English of the manuscript.